

# On the role of the South Pacific subtropical high at the onset of El Niño events

Youjia ZOU[1], Xiangying XI[2]*

[1]Department of Meteorology and Oceanography, Shanghai Maritime University, Shanghai 201306, China.

[2]Wuhan University of Technology, 122 Luoshi Road, Wuhan 430070, China.

Youjia ZOU: marscar@126.com

*Xiangying XI: xxy9898@aliyun.com

*Correspondence to: xxy9898@aliyun.com

## Abstract

**Previous studies have suggested that an eastward propagation of the warm pool in the western Pacific during El Niño events may be induced by a weakening of the easterly Trade Winds (Alexander et al. 2002; Bjerknes 1969). However, the dynamic mechanism of the Trade Winds weakening is not well understood. Here we use a model and other published proxy records to demonstrate that the anomalous southward shift of the south Pacific subtropical high (SPSH) may play a crucial role at the onset of El Niño events. By analyzing the relationship between the Trade Winds, the Equatorial Currents, the Eastern Boundary Currents and the SPSH, we find that an anomalous southward shift of the SPSH can result in a weakening of the SE Trade Winds and a southward intrusion of the NE Trade Winds, leading to a southward migration of the Trade Wind-induced Equatorial Currents, including the Equatorial Countercurrent (from ~5°-8°N to ~0°). The warm pool in the western equatorial Pacific is**



**therefore forced to propagate eastward by the enhanced Equatorial**
**Countercurrent and, thus, a warm phase in the central or the eastern equatorial**
**Pacific. Moreover, the equatorward upwelling in the eastern South Pacific,**
**usually recurring along the equator, shifts southward along with the SPSH, in**
**turn diverts towards the west at ~15 ̊S to feed the westward South Equatorial**
**Currents, resulting in a failure of cooling sea surface in the eastern tropical**
**Pacific, thus a flattening of the thermocline. The model experiments indicate that**
**the meridional position and intensity of the Equatorial Countercurrent in the**
**Pacific are some of the determining factors in giving rise to El Niño diversity,**
**suggesting that there should be more frequent warm events due to a meridional**
**expansion of the warm pool under global warming.**
**Key Words**
El Nino; subtropical high; southward shift; weakening of the trade winds; southward
shift of the equatorial currents; southward shift of the upwelling;


**Introduction**
The El Niño phenomenon, characterized by anomalous Trade Winds and sea surface
temperatures (SSTs) in the tropical Pacific (Bjerknes 1969; Ramesh & Murtugudde
2013), is considered to have global implications with costly consequences. Presently
there is a general agreement in the fields of the atmospheric and oceanic science that
the warm pool (SSTs greater than about 29 ̊C) in the western Pacific propagating





eastward along the equator is induced by the weakening of the Trade Winds
(McPhaden 1999). This picture, however, leaves open the question of why and how
the Trade Winds weaken. Despite a variety of mechanisms being proposed (Bjerknes
1969; Wyrtki 1975; Oldenborgh 2000), there is no scientific consensus on how the
Trade Winds slacken or even reverse. The apparent absence of a super warm phase in
2014 that was expected by many models implies that we may still not understand
some fundamental aspects of the system. Over the past decades, investigations into
the tropical Pacific's role at the onset of El Niño events mainly focused on the SST
anomalies (that is, deviations from climatological norms) (Rasmussen & Carpenter
1982), recharge/discharge of equatorial upper-ocean heat content (Meinen &
McPhaden 2000) and westerly wind bursts (Lengaigne 2004; Fedorov et al. 2014).
Recently, the westward equatorial currents were found to be enhanced during La Niña
but distinctly reversed during extreme El Niño events (Santoso et al. 2013). Our
investigations show that all the above are likely to be directly associated with the
South Pacific Subtropical High system (hereafter referred to as SPSH), which has
potential (because the Trade Winds, the Westerlies and the Eastern Boundary
Currents are all mainly associated with the SPSH and further develop with it in
position and intensity) to dominate climate change in South Pacific region by creating
significant impacts on the Trade Winds, precipitation patterns and ocean circulations
(**Figs.1a-b**). We therefore hypothesize that there may be a critical role played by the
SPSH at the onset of El Niño events.




Each year, the South Pacific experiences a seasonal cycle with a northward/southward
shift of the subtropical high (SH) in austral winter/summer from ~16°S to ~35°S
(Reid et al. 1958). It is generally accepted that the seasonal migrations of the SPSH
cannot make significant impacts on the Trade Winds, the Equatorial Currents and the
Eastern Boundary Currents due to its nonlinear mechanism. Nevertheless, above
seasonal migrations may be disturbed when the South Pacific undergoes a
perturbation from external forcings, for example, an insolation weakening, leading to
an anomalous displacement of the SPSH (**Figs.1a-b**). Evidence shows that the SH is
sensitive to the external forcings (Reid et al. 1958).

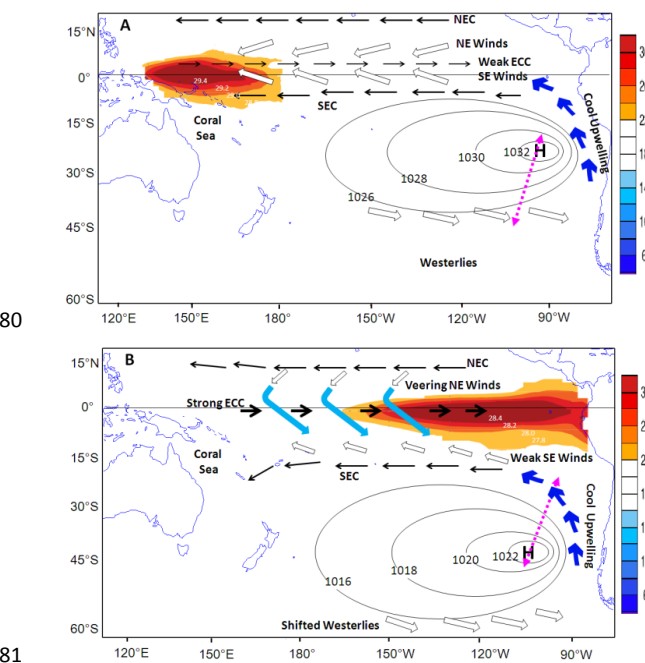



**Fig 1**.**Comparison between normal and El Niño conditions. A,** normal condition with strong trade
winds, weak ECC and intense upwelling recurving along the equator. Warm waters brought by the
weak ECC mix with the strong upwelling in the central equatorial Pacific. **B,** El Niño condition with a
southward shifted SPSH (along with the SE trade winds, equatorial currents and westerlies), veered NE
trade winds, strong ECC and weak upwelling deflecting to the west at ~15°S. Warmer waters brought
by the strong ECC mix with the weak upwelling in the eastern South Pacific, leading to SSTs





anomalies. The dashed arrows in pink color denote the approximate trajectory of the SPSH shift. The
solid light blue arrows denote veering Trade Winds. The hollow arrows, solid black arrows and solid
dark blue arrows represent the climatological winds, currents and cool upwelling, respectively. The
Trade Winds are symmetric about the wind equator (about ~5 °N-8 °N) in normal condition, rather than
the geographic equator.

An anomalous southward migration of the SPSH can result in a weakening of the SE
Trade Winds and an enhancement of the Equatorial Countercurrent (ECC),
concurrently allowing for a southward incursion of the NE Trade Winds (**Fig.1b**). As
the Trade Wind system shifts southerly, so do the Trade Wind-induced equatorial
currents. The Pacific ECC (hereafter referred to as ECC), the strongest (more than 20
Sv) compared with its counterparts (Yu et al. 2000), residing between the North
Equatorial Current (NEC) and the South Equatorial Current (SEC), with its mean axis
usually around 5 °N in winter and 8 °N in summer (Yu et al. 2000; Tomczak &
Godfrey 2003), migrates to about 0 ° or more south in response to the southward shifts
of the Trade Wind system, advecting the giant pool of the warm waters eastward
along the equator (**Fig.1b**). In essence, Wyrtki postulated in 1973 that an unusually
strong ECC in the western Pacific would lead to an anomalous accumulation of the
warm water in the eastern equatorial Pacific and, thus, El Niño event (Wyrtki 1973),
but his suggestion has long been overlooked due to lack of plausible mechanisms and
a failure of explaining why the warm pool propagates along the equator rather than
along the ~5 °N or ~8 °N of latitude.

In addition, a southward migration of the NE Trade Winds can result in a veering of
the NE Trade Winds from northeast to northwesterly or westerly under the influence





of the Coriolis force after the NE Trade Winds cross the equator, further amplifying
the intensifying of the ECC (**Fig.1b**). Meanwhile, a relaxation of the SEC in response
to a weakening of the easterly winds is liable to lead to less build-up of heat content in
the western tropical Pacific but more heat is retained in the central and the eastern
tropical Pacific (McPhaden 1999). The net result of all the above is a reversal of the
Walker Circulation, creating westerly winds in the western tropical Pacific and
intensifying the eastward ECC, thus establishing a positive feedback. It is noteworthy
that the source of the ECC has changed from the warm water to warmer water with
southward shifts of its main axis from ~5°N-8°N to ~0°, fueling the reversal of the
Walker Circulation and the warming in the central and the eastern equatorial Pacific.
Moreover, the upwelling in the eastern South Pacific, which usually recurves along
the equator, shifts southward along with the Trade Winds/SPSH, in turn diverts
towards the west at ~15°S during El Niño events to feed the westward SEC
(**Figs.1a-b**), resulting in a failure of cooling sea surface in the eastern tropical Pacific,
thus a flattening of the thermocline.

**Simulating El Niño events**
To test whether the SPSH acts as a possible trigger at the onset of El Niño events, we
carry out simulation experiments in which we examine the response of SSTs in the
tropical Pacific to the observed location and intensity of the SPSH added to a
comprehensive climate GCM, HadOPA, which couples the OPA (ocean model) and
HadAM3 (atmospheric model) through OASIS 2.4 (Lengaigne et al. 2004 & 2006).



(For details of the model description, see Methods). We slightly modify this model
and assume that the surface wind stress anomaly and the ECC anomaly are a function
of the position and strength of the SPSH (see Methods). When the surface wind stress
and the meridional position of the ECC vary by artificially tuning the position and
strength of the SPSH as a perturbation, the interannual oscillations with SST
anomalies retain little change at the initial stage due to its nonlinear effects but start to
surge and become highly irregular as the SPSH continuously moves southward in
early spring with a gradual weakening. When further perturbation is imposed in late
spring, the model produces a broad continuum of El Niño events subsequently in
position ranging from the dateline to the eastern tropical Pacific (Extended Data
**Figs.3a-c**). A strong El Niño event occurs in winter in the eastern tropical Pacific as
an intense southward shift of the SPSH superimposed on the seasonal cycle takes
place (Extended Data **Fig.3c** and Methods). However, a relative weak El Niño event
appears in summer around the dateline when a weak southward migration of the
SPSH occurs (Extended Data **Fig.3a** and Methods). We run this model by changing
the meridional position anomalies of the SPSH ($\Delta\varphi_{spsh}$) to ~+10° and ~+12° of
latitude (observed location anomalies), respectively, with a gradual weakening, to
simulate the El Niño episodes in 1982 and 1997. As expected, the warm events
quickly develop into extreme EP El Niño events with SST anomalies in Niño3
exceeding 3.8℃ and 4.2℃, respectively, consistent with the observations (**Figs.2a-b**).
(data available online at http://www.cpc.noaa.gov/data/indices)





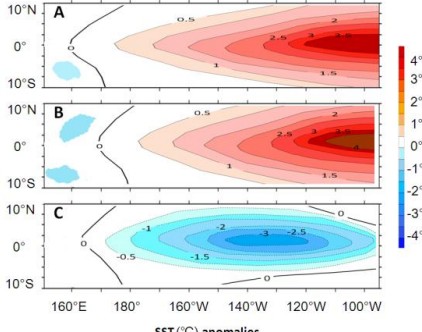


**Fig 2.Simulated El Niño and La Nina. A,** a strong El Niño in the eastern tropical Pacific as the $\Delta\varphi_{spsh}$
is ~+10 ° of latitude. **B,** a stronger El Niño in the eastern tropical Pacific as the $\Delta\varphi_{spsh}$ is ~+12 ° of
latitude. **C,** a La Niña near the eastern tropical Pacific as the $\Delta\varphi_{spsh}$ is ~-4 ° of latitude with zonal
position anomalies ($\Delta\lambda_{spsh}$) ~+7 °of longitude (an eastward anomaly is positive).

### Simulating La Niña

Similar model experiments have been done to simulate the La Niña episode in 2008
by moving the SPSH northerly. The simulation experiments indicate that the warm
phase in the eastern tropical Pacific subsequently evolves into a cold phase in late
summer the next year as the $\Delta\varphi_{spsh}$ is about ~-4 ° of latitude (a northward anomaly is
negative) and the zonal position anomalies ($\Delta\lambda_{spsh}$) are +7 ° of longitude (an eastward
anomaly is positive), with 2℃-3℃ cooling of SST anomalies in Niño3 region
(**Fig.2c**), reasonably consistent with the observed records. Northward shifts of the
SPSH can enhance the SE Trade Winds and the SEC, weaken the ECC and push the
SEC and ECC northwards, leading to a westward shift of the warm pool and the
atmospheric convection in the equatorial Pacific. Whether an El Niño event is
followed by a La Niña principally rests with the $\Delta\varphi_{spsh}$ and the upwelling feedbacks
which are mainly determined by the southerly onshore winds in the eastern part of the
SPSH (Rollenbeck et al. 2015). The upwelling feedbacks tend to be stronger when the



zonal pressure gradients in the eastern part of the SPSH are steeper (more dense
isolines) and the center position of the SPSH from South American coast is more
favorable (**Fig.6c**). The simulation experiments suggest that an approximately
NNE-SSW oriented trajectory of the SPSH shift is conducive to more steep zonal
pressure gradients in the eastern part of the SPSH when the SPSH shifts northerly
according to the theory of fluid mechanics and Bernoulli's theorem, generating more
intense upwelling (the shifting trajectory of the SPSH centre and coast line produce a
duct-like passage for the southerly onshore winds with a narrow opening in the north
and a relatively wide opening in the south) (**Fig.1b**). In addition, a more northerly
location of the SPSH tends to bring the upwelling to the right position (around equator)
in the eastern equatorial Pacific, favoring a La Niña. However, in realistic regimes,
the transit from a warm phase to a cold phase may be slightly different from that
created by our theoretical models, possibly involving a more complex process, such as
an oscillating, a pause or a prolonged evolution, etc.

**Simulating ECC anomalies**
The experiment in simulating the response of the meridional position anomalies of the
ECC ($\Delta\varphi_{ecc}$) to the $\Delta\varphi_{spsh}$ indicates that the $\Delta\varphi_{ecc}$ nonlinearly corresponds to the $\Delta\varphi_{spsh}$
(**Figs.3a-d, extended data Fig.5**), suggesting that the meridional position and
intensity of the ECC are some of the determining factors in giving rise to El Niño
diversity. A strong ECC in an appropriate meridional position (around the equator)
tends to advect more warm waters to the eastern equatorial Pacific and produce an




extreme EP El Niño (a more southerly position of the ECC in winter is easier to be
pulled down to the equator, offering a sufficient explanation for extreme El Niño
events always occurring in winter) (**Fig.3d**), implying that there should be more
frequent warm events due to a meridional expansion of the warm pool in the western
equatorial Pacific and an acceleration of the Hadley Cell (poleward shifts of the
descending points) under global warming (Cravatte et al. 2009).

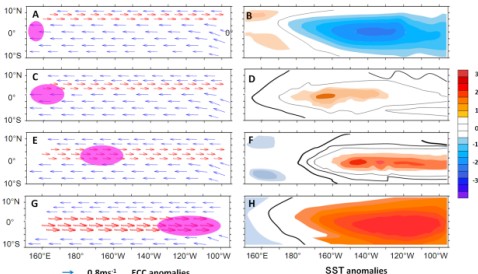


**Fig 3**.**Simulated ECC anomalies and corresponding SST anomalies. A** and **B,** a weak ECC in La
Nina condition and corresponding SST anomalies. **C** and **D,** a moderate and southward shifted ECC as
the $\Delta\varphi_{spsh}$ is ~+7° of latitude and corresponding SST anomalies. **E** and **F,** a strong, broad and
southward shifted ECC as the $\Delta\varphi_{spsh}$ is ~+10° of latitude and corresponding SST anomalies. **G and H,**
a stronger, broader and southward shifted ECC as the $\Delta\varphi_{spsh}$ is ~ +11° of latitude and corresponding
SST anomalies. The warm pool is brought to the central or eastern equatorial Pacific by the southward
shifted ECC. The red and blue arrows represent the ECC and the NEC/SEC, respectively. The shading
area denotes the warm pool on the left panels.

**Discussion**
A key question being debated for long time is whether the southward migration of the
SPSH is a passive response to El Niño events or is a driver to El Niños. Traditionally,
the southward migrations of the SPSH are thought by some authors to be a result of El
Niños (McPhaden, 1999; Meinen & McPhaden 2000; Oldenborgh 2000). In contrast,
our investigation reveals that a southward shift of the SPSH is not a passive response
to El Niño events but is driving El Niño events. This reasoning is based on the





asymmetric response of the North Pacific Subtropical High (NPSH) and the SPSH to
the eastward SSTs anomalies (the equatorward displacements of the NPSH were
observed in winter during El Niño events), inconsistent with that both the NPSH and
SPSH should be synchronously affected by the eastward SSTs anomalies if the
southward migration of the SPSH were the result of El Niño events. Besides, the
SPSH is a large-scale permanent pressure system produced by the global general
circulation of the atmosphere, rather than by an individual equatorial low pressure belt
in the equatorial Pacific. Therefore the SPSH is not likely to be driven by a regional
system, such as the local warming/cooling in the equatorial Pacific. Furthermore, the
anomalous migrations of the SH have also been identified in other oceans (Zou et al.

231  2017).


The strong support for the southward shifts of the SPSH not being forced by the
eastward warm pool during El Niño events comes from two independent
investigations into proxy records and experiments throughout the Holocene. The
fluctuations of the iron concentrations, which are thought to reflect the precipitation
patterns in southern Chile, intimately linking with the westerlies (Lamy et al. 2001),
are qualitatively consistent with the periods of ENSO (Moy et al. 2002) in the past 8
kyr (**Extended Data Fig.4**), suggesting southward displacements of the westerlies in
the South Pacific during El Niño events, thus implying the concurrent shifts of the
SPSH (because the westerlies are mainly associated with the SPSH and further
develop with it in position and intensity). The solar sensitivity experiments with a



comprehensive global climate model indicate that the southward migrations of the
westerlies are in line with the variations of solar forcing (Varma et al. 2001), implying
that the southward shifts of the SPSH during El Niño events are likely attributed to
solar activity, rather than El Niño itself, and further supporting our hypothesis.
Furthermore, the timing of the southward displacements of the westerlies was
concurrent with that of the strengthening of the ECC, also suggesting a role of the
southward displacements of the SPSH at the onset of El Niño events.

Another theory worth noting is the "westerly wind burst" which is recently suggested
to be a possible trigger of El Niño events (Lengaigne et al. 2004; Fedorov et al. 2014;
Menkes et al. 2014). These westerly winds are thought to be a manifestation of the
Madden-Julian Oscillation (MJO) which originates over the Indian Ocean,with a 30
to 60-day period (Madden & Julian 1972). However, McPhaden (1999) argued that
the episodic westerly wind forcing is not a necessary condition for the development of
El Niño events because such forcing can be seen during non–El Niño years, and many
coupled ocean-atmosphere models also simulate ENSO-like variability without it
(McPhaden & Yu 1999). Our investigation shows that the observed "westerly winds"
are to some extent ascribed to the veered NE Trade Winds after crossing the equator
(subsequently becoming northwesterly or westerly winds), constituting the lower limb
of the reversed Walker Circulation in the western-central tropical Pacific during El
Niño events. In essence, the Trade Winds in the equatorial Pacific, in contrast to that
in the equatorial Atlantic, are not symmetric about the geographic equator, but about





the ~5 °N-~8 °N of latitude (climatological mean position, a.k.a "wind equator"). This
region (0 °-~5 °N in the middle of the Pacific) is actually dominated by the SE Trade
Winds with frequency of 40%-50% (Sailing Directions, 2013) rather than by the NE
Trade Winds intuitively thought. This is confirmed by Routeing Chart 5127 (UK
hydrographic office, 2012), suggesting that the NE Trade Winds start to deflect to
northwesterly or westerly (mean extending latitudinally from ~5 °N to ~10 °S) once
beyond the "wind equator". This explains why the westerly winds can be seen in
north of the geographic equator. Occasionally the maximum northern boundary of the
westerly winds can reach ~10 °N under the influence of effects of the entrainment. The
ellipse-shaped structure of the SPSH may account for the westerly winds occurring in
the western equatorial Pacific first (gradually towards the central Pacific) as the SPSH
migrates southward, consistent with the satellite observations indicating reversed
Trade Winds mainly confined in the western and central Pacific during El Niño events
(**Extended Data Fig.6**). The model experiments indicate that although the
MJO-related westerly wind forcing is not a sufficient condition for the El Niño onset,
it can amplify the veered NE Trade Winds if it occurs on time, reinforcing the
reversed Walker Circulation and the ECC and, thus, promoting the El Niño-like states
to evolve to El Niño events (**Fig.3d**). This explains why every warm event during the
past 50 years was always preceded by the westerly winds (Eisenman et al. 2005).

The superposition of the MJO-related westerly winds onto the veered NE Trade
Winds may contribute to surface water convergence (Chen et al. 2015), promote



eastward downwelling equatorial Kelvin waves that create warming phase in the
eastern tropical Pacific (McPhaden & Yu 1999), advect the warm pool eastwards and
push the ECC southwards and eastwards (Picaut et al. 1997), leading to a broader and
stronger ECC (**Figs.3b-d**), consistent with the satellite observations (**Fig.4a-d**). In
contrast to some previous studies (Lengaigne et al. 2004; Fedorov et al. 2014), we
argue that the eastward propagation of the warm pool in the western equatorial Pacific
is likely to be forced mainly by the enhanced and southward shifted ECC rather than
by the episodic westerly winds because those westerly winds were observed to be
sporadic and not strong enough (Beaufort Scale 5 or less) even if in the most
pronounced event in 1997 according to the satellite observations (**Extended Data**
**Fig.6**), but these winds may help the eastward development of the warm pool. The
likelihood of the westward equatorial currents (SEC) being totally reversed by the
sporadic and weak westerly winds is considered low. The newly-discovered "reversed
Equatorial Currents" during extreme El Niño events by Santoso et al. (2013), is most
likely to be the southward shifted ECC when combined with other evidence from
Wyrtki (1973), Lamy et al.(2001) and Varma et al.(2011). This is also confirmed by
the satellite observations with an absence of the ECC in previous latitudes (~5°N-8°N)
and an emergence of a new eastward equatorial current around the equator during El
Niño events (**Fig.4a-d**). The fact that the westward transport of the SEC entering the
Coral Sea (in northeast of Australia, **Fig.1a-b**) increases during El Niño events and
decreases during La Niña (Kessler & Cravatte 2013) is suggestive of the meridional
shifts of the SEC (the climatologically strongest SEC meridionally ranging from ~2°N



to ~6 °S, then gradually weakening towards the south (Yu et al. 2000; Tomczak &
Godfrey 2003). Also see Routeing Chart 5127 published by UK Hydrographic Office
in 2012), further confirming our speculation.

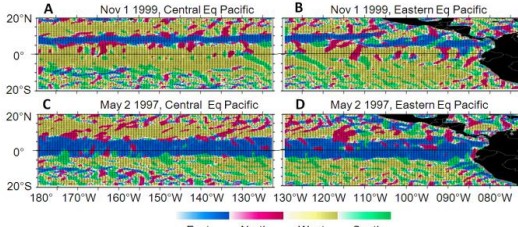


**Fig 4. Surface Current derived from satellite observations**. **A** and **B,** the positions and directions of
the surface currents in the central and eastern equatorial Pacific, respectively, during normal condition
(Nov 1, 1999). **C** and **D,** the positions and directions of the surface currents in the central and eastern
equatorial Pacific, respectively, during El Niño condition (Nov 1, 1997).   A broader and southward
shifted ECC can be seen around the equator. Different colors denote different directions of the surface
currents. http://www.oceanmotion.org/html/resources/oscar.htm.

**Simulating the tropical wind anomalies and upwelling**
To further examine the response of the tropical wind anomalies to the position
anomalies of the SPSH, we run the model by altering the meridional position of the
SPSH alone (Methods). Over the tropical Pacific, the model simulation shows that the
tropical wind anomalies closely track the variations of the SPSH. As anticipated, the
tropical wind anomalies are not evident by changing the zonal position of the SPSH
alone (**Fig.5**). In 1997, the center of the SPSH was observed to shift from 27 °S in
May to 36 °S in August, to 45 °S in November, all at about 77 °W-80 °W, with zonal
wind anomalies at lat 0 °/long 150 °E from 1ms$^{-1}$ in May to 5ms$^{-1}$ in August, to 7.5ms$^{-1}$
in November, respectively (McPhaden 1999). The superposition of the El
Niño-related southward shifts of the SPSH onto the seasonal cycle makes the average



speed of the SPSH moving nearly 50 percent faster than usual, serving as an
alternative precursor for the initial development of the event. This offers the scientists
new insights into monitoring and prediction of the El Niño onset.

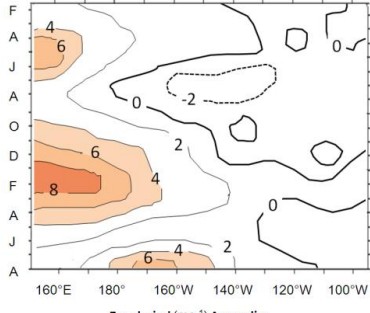


**Fig 5**.**The response of the zonal wind anomalies in the equatorial tropical Pacific to the**
**meridional position anomalies of the SPSH.** The zonal wind anomalies (between 5°N and 5°S) as the
$\Delta\varphi_{spsh}$ is ~+7° of latitude during Mar-Jun and the zonal wind anomalies as the $\Delta\varphi_{spsh}$ is ~+10° of
latitude during Nov-Apr.

Similar model experiment has been executed for validating the response of the
upwelling to the position anomalies of the SPSH, indicating that the upwelling
sharply follows the alterations of the SPSH in zonal position, with the meridional
position playing a negligible role (**Figs.6a-c**). The model simulations suggest that the
tropical wind anomalies are affected primarily by the meridional position of the SPSH
through varying the surface wind stress while the intensity of the upwelling is mainly
influenced by the zonal location of the SPSH, with the meridional position and the
strength of the SPSH playing a secondary role, consistent with the previous study on
the NPSH (Cheshire & Thurow 2013). The zonal pressure gradients near the center of
the SPSH are small but it can be huge with strong southerly onshore winds in the
eastern part of the SPSH (the densest isolines, **Figs.6b-c**), generating intense





upwelling along South American coast if a right zonal position of the SPSH is set.

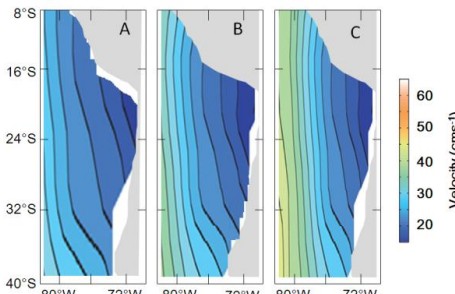


**Fig 6.The response of the upwelling feedbacks to the zonal position anomalies of the SPSH. A,** the
upwelling feedbacks (surface velocity) in normal condition. **B**, the upwelling feedbacks as the $\Delta\lambda_{spsh}$ is
+6 ° of longitude, **C**, the upwelling feedbacks as the $\Delta\lambda_{spsh}$ is +8 ° of longitude (the meridional position
and the intensity of the SPSH remain unchanged). The smaller the spacing, the stronger the upwelling
feedbacks.

**Conclusion**
The model experiments suggest that the SPSH may play a critical role at the onset of
El Niño events. Further development of El Niño events (diversity) is likely to be
influenced by the subsequent air-sea interactions and the interplay between the
eastward warm pool in the western tropical Pacific and the unstable mixing state of
warm and cold waters in the central or the eastern tropical Pacific. This does not,
however, disparage other drivers which may also play a role at El Niño onset.
Understanding the role of the SPSH at the onset of El Niño events is important not
only because it is capable of fully reconciling the divergent views of El Niño's origin
but also because it exhibits a more plausible explanation of El Niño/La Niña. The
apparent lack of real-time forecasting and long-term predictability of El Niño at the
current stage implies that we have some way to go in fully understanding the real
physical mechanisms of the El Niño/La Niña phenomenon. It is believed that our new



findings can better shed light on the role of the SPSH in the genesis of El Niño and
may lead to more accurate predictions for a longer period in the future.

**Acknowledgements** authors sincerely acknowledge Trenberth K.E., who is currently
working in NCAR, USA, for providing valuable suggestions in the contributors of El
Niño onset.

**Author Contributions** Both authors contributed equally to this work.
Zou collected all data, prepared the manuscript and figures, and performed the
analysis. Xi was responsible for data collection, laboratory efforts and contributed to
the computer programming and the model simulating. Both authors discussed the
results and provided inputs to the paper.

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
