# Peer review of "On the role of the south Pacific subtropical high at the onset of El Niño events"

_Atmospheric Chemistry and Physics, 2018_

## Referee Comment (RC1) · Anonymous Referee #1 · 28 May 2018

This paper hypothesizes that meridional (southward) migration of the south Pacific subtropical high (SPSH) is important for the onset of El Nino events by causing a weakening of the trade winds and anomalous eastward flow along the equator. The hypothesis is tested using a series of coupled model experiments in which a meridional migration of the SPSH is imposed on a climatological mean state. El Nino ensues with the SPSH is displaced southward and La Nina ensues when the SPSH is displaced northwards.

The paper appears to discover something that is already known, namely that variations in the SPSH affect the evolution of ENSO. The clearest indicator of this is the very high correlation between the Southern Oscillation Index (SOI) and oceanic indices for ENSO, e.g. the NINO3.4 SST index. The SOI is the difference between surface pressure at Darwin and surface pressure at Tahiti, the latter of which is located

in the SPSH. Low SOI is indicative of weakened southeasterly trade winds and high SOI of strengthened trade winds.

The authors argue that the North Equatorial Countercurrent (labeled ECC in the paper) migrates to the equator during El Nino. The eastward flow along the equator during El Nino is not the ECC but a reversal of the South Equatorial Current (SEC) that results from a weakening the trade winds. The anomalous eastward currents in the SEC result from the fact that when the winds weaken, the unbalanced zonal pressure gradient causes an eastward acceleration of the flow along the equator.

The authors also seem to discount the role that westerly wind burst (WWB) forcing plays in the evolution of El Nino because WWBs occur in non-El Nino years as well as El Nino years. The effectiveness of WWBs however is conditioned on whether there is a prior build up of heat content along the equator which the authors seem not to understand. Almost all studies that have examined the role of these episodic wind forcing events conclude they play a very important role in the ENSO cycle helping to initiate and amplify El Nino warming.

I found this paper more confusing than helpful and recommend rejection.

---

## Author Comment (AC1) · 29 May 2018

Dear colleague, I'm very delighted to provide some explanations for your comments: 1. SOI and NINO3.4 SST index both are indices of ENSO. Traditional thoughts consider that a southward shift of the SPSH is a result of the Ei NINO envents, but we think it is a possible diver of Ei NiNO events. This is the substantial difference.

2. Based on the comments of Anonymous Referee #1, the eastward current along the equator during Ei Nino events is a reversed SEC. If it were true, it should have two branches of the eastward currents, one (ECC) along 5-8 Latitudes, the other along the equator, inconsistent with the satellite observations.

3. it is well known that the broad (in meridional extent) westward SEC is induced by the

easterly winds. To reverse the westward SEC from westward to eastward needs long-term, steady and persistent westerly winds. The sporadic, short-duration and weak westerly winds observed during Ei Nino events is considered insufficient.

4. If the eastward current along the equator during EiNino envents were induced by the unbalanced zonal pressure gradient (the westerly winds), the relatively narrow eastward currernt (in meridional extent) along the equator could not be explained by the broad westerly winds observed in south of the equator.

---

## Author Comment (AC2) · 29 May 2018

Dear Colleague,

We would like to talk more about your comments:

1. the westerly wind fetch (in meridional extent) in south of equator during El Nino events is broad, extending to about 15S (Fedorov et al., 2014). If the eastward current along the equator were induced by these westerly winds, it should be as wide as the meridional wind fetch, inconsistent with the satellite observations (a relatively narrow branch).

2. southward shift of the ECC along with the Trade winds is plausible and is easier than a reverse of the SEC.

3. the coincidence of an absence of the ECC in previous latitudes (5N∼8N) and an emergence of a new eastward current along the equator is likely to be suggestive of a southward shift of the ECC.

wish it helpful for your further understanding

Best Regards

---

## Short Comment (SC1) · 13 Aug 2018

This manuscript offers a brand new insight in the field, different from previous descriptions. The evolution of the ECC position, however, still remain unclear. The coincidence of an absence of the ECC in previous latitudes (5N–8N) and an emergence of a new eastward current along the equator can not fully corroborate that the eastward current is the southward shifted ECC. More details are needed.

---

## Author Comment (AC3) · 14 Aug 2018

Dear colleague, sorry for lack of much more details because of space limited. We are currenty doing a work showing the evolutions of the ECC in latitude in time in another paper, indicating that the eastward current along the equtor is the shifted ECC, rather than a reversed SEC. Regretably, it is not possible to add more in this manuscript.

Your understanding is much appreciated!

Best Regards

Authors

2018.

---

## Referee Comment (RC2) · Anonymous Referee #3 · 23 Aug 2018

The paper studies the impact that a southward migration of the south Pacifical high might have on the onset of El Niño events, using a coarse-resolution coupled ocean-atmosphere model and some proxy data. While the question is potentially interesting I find the manuscript confusing and the proposed mechanism not very convincing. Therefor I do not recommend the paper for publication.

Some comments:

1) A central argument of the paper is that the North Equatorial Countercurrent (referred to as ECC by the authors) moves onto the Equator in response to a southward shift of the SPSH. This is most certainly not the case in reality (and the authors also do not provide convincing evidence that it is the case in their model). As for observed and simulated variations of the NECC on interannual time scales, I recommend the

paper by Hsin and Qiu (2012) and the references therein. They indicate that the NECC indeed does move southward during Eastern Pacific El Niño events but not by more than about one degree.

2) The authors spend quite an amount of text and supplementary figures on the fact that the trade winds do, on average, not converge on the Equator but north of it. This is standard textbook knowledge and by the way also the case in the Atlantic Ocean, in contrast to what the authors claim.

3) The manuscript is rather hard to follow as it is not very well structured (main results are already discussed in the introduction, the methodology is completely missing from the main manuscript), the figures are tiny and the figure captions do not provide the necessary information. For example, in Fig. 3 it is not clear what depth or density range is shown and what time period is considered.

Reference: Hsin, Y.-C., and B. Qiu (2012), The impact of Eastern-Pacific versus Central-Pacific El Niños on the North Equatorial Countercurrent in the Pacific Ocean , J. Geophys. Res., 117, C11017, doi:10.1029/2012JC008362.

---

## Author Comment (AC4) · 24 Aug 2018

Dear Anonymous Referee #3, Many thanks for your comments.

Comments from Anonymous Referee #3: 1) A central argument of the paper is that the North Equatorial Countercurrent (referred to as ECC by the authors) moves onto the Equator in response to a southward shift of the SPSH. This is most certainly not the case in reality (and the authors also do not provide convincing evidence that it is the case in their model). As for observed and simulated variations of the NECC on interannual time scales, I recommend the paper by Hsin and Qiu (2012) and the references therein. They indicate that the NECC indeed does move southward during Eastern Pacific El Niño events but not by more than about one degree.

[Figure]

Response to Anonymous Referee #3 (1) You provided a reference for the ECC (or NECC) position not significant southward shifting during El Ninos against our view. You know that the paper written by Hsin and Qiu (2012) might be right in 2012, but this doesn't guarantee that it is always right with the developemnt of evidence obtained by the state-of-the-art techniques. Here we offer a more clear picture for the evolution process of the NECC position in latitude (see the figure in attachment).

The satellite observations in situ show that the NECC in Nov 1996 was in the normal position 5∼8°N (blue color) and remained in this position in Dec 1996, Jan and Feb 1997 but with considerably anomalous southward currents (green color, likely in an oceanic disturbed state). Subsequently the NECC passed latitude of 5°N in Mar. The southern edge of the NECC clearly shifted southward beyond the equator in Apr, with a stronger intensity and a broader range in latitude possibly due to northwest or westerly winds. The NECC finally fixed at latitudes of 4°S∼6°N in May, representing the start of an El Niño, consistent with the beginning time of El Niño event in1997. The NECC stayed at this position for nearly 13 months until the end of 1997 El Niño. The evolutions of the NECC position in latitude with time lend sufficient support to the notion of a southward shifted NECC during El Niños.

The evolution process can be clearly seen in a set of daily change maps of the NECC position . But it is not possible for us to arrange so many figures here.

The satellite observational records clearly show that the NECC at latitudes of 5∼8°N disappeared and a new eastward current along the equator appeared at the same time in 1997. The synchronicity of an absence of the NECC in the normal latitudes and an emergence of a new eastward current along the equator is indicative of a southward shifted NECC.

please note that this is the satellite observation provided by NOAA, not an artificial work.

Comments from Anonymous Referee #3: 2) The authors spend quite an amount of

text and supplementary figures on the fact that the trade winds do, on average, not converge on the Equator but north of it. This is standard textbook knowledge and by the way also the case in the Atlantic Ocean, in contrast to what the authors claim.

Response to Anonymous Referee #3 (2) We describe the trade winds symmetry about 5∼8°N rather than 0° in order to explain why the northwest winds can be seen in north of the equator. Although this exists in the Atlantic, the amplitude is much smaller than in the Pacific.

Comments from Anonymous Referee #3: 3) The manuscript is rather hard to follow as it is not very well structured (main results are already discussed in the introduction, the methodology is completely missing from the main manuscript), the figures are tiny and the figure captions do not provide the necessary information. For example, in Fig. 3 it is not clear what depth or density range is shown and what time period is considered.

Response to Anonymous Referee #3 (3) We would like to further improve the quality of the figures and figure captions as well. The figures with sufficient resolution can be zoomed in as big as you want. Fig.3 shows the surface sea conditions in the equatiroal Pacific at any period, representing general conditions in La Nina and various El Nino events. Because the GCM is employed from the other authors, therefore we suggest that it is not necessary to repeat the details of the methodology in the main text, instead, with references only.

If it is declined, the audience in this field would lose an opportunity to see a new view different from the traditional one, but this new view may be right with the development of the new technology and methods.

You see that this paper has been cited by 16 authors during the duscussion stage, and this figure may increase quickly in a short period after formally pblished, representing its potential interests to the audience in this field.
* * *
[Figure]

2018.

[Figure]

**Fig. 1.** Evolution of the NECC position in latitude